# Analysis of the Genomes and Adaptive Traits of *Skermanella cutis* sp. nov., a Human Skin Isolate, and the Type Strains *Skermanella rosea* and *Skermanella mucosa*

**DOI:** 10.3390/microorganisms13010094

**Published:** 2025-01-06

**Authors:** Yujin Choi, Munkhtsatsral Ganzorig, Kyoung Lee

**Affiliations:** Department of Bio Health Science, Changwon National University, Changwon 51140, Gyeongnam, Republic of Korea; dwls0863@naver.com (Y.C.); gmunkhtsatsral1@gmail.com (M.G.)

**Keywords:** *Skermanella*, human skin, nitrogen fixation, denitrification

## Abstract

The genus *Skermanella* comprises important soil bacteria that are often associated with the crop rhizospheres, but its physiological traits remain poorly understood. This study characterizes *Skermanella* sp. TT6^T^, isolated from human skin, with a focus on its metabolic and environmental adaptations. Genome sequencing and phylogenomic analyses revealed that the strain TT6^T^ is most closely related to *S. rosea* M1^T^, with average nucleotide identity and digital DNA–DNA hybridization values of 94.14% (±0.5%) and 64.7%, respectively. Comparative genomic analysis showed that the strains TT6^T^, *S. rosea* M1^T^ and *S. mucosa* 8-14-6^T^ share the Calvin cycle, and possess photosynthetic genes associated with the purple bacteria-type photosystem II. The strains TT6^T^ and *S. rosea* M1^T^ exhibited growth in a nitrogen-free medium under microaerobic conditions, which were generated in test tubes containing 0.1% soft agar. Under these conditions, with nitrate as a nitrogen source, *S. rosea* M1^T^ formed gases, indicating denitrification. Strain TT6^T^ also contains gene clusters involved in trehalose and carotenoid biosynthesis, along with salt-dependent colony morphology changes, highlighting its adaptive versatility. Genomic analyses further identified pathways related to hydrogenase and sulfur oxidation. Phenotypic and chemotaxonomic traits of strain TT6^T^ were also compared with closely related type strains, confirming its genotypic and phenotypic distinctiveness. The new species, *Skermanella cutis* sp. nov., is proposed, with TT6^T^ (=KCTC 82306^T^ = JCM 34945^T^) as the type strain. This study underscores the agricultural and ecological significance of the genus *Skermanella*.

## 1. Introduction

The genus *Skermanella*, belonging to the family *Azospirillaceae*, within the class *Alphaproteobacteria*, was first proposed by the introduction of *S. parooensis* ACM 2042^T^ as the type species of the genus [1]. The family *Azospirillaceae* was recently revised, based on genome sequence homologies, and the genus *Skermanella* is phylogenetically closely related to the genus *Azospirillum*, which comprises diazotrophic soil bacteria [2]. A metagenomic analysis of soil samples and a cDNA analysis of the *nifH* gene, which encodes the reductase component of Mo-nitrogenase, have shown that *Skermanella* strains are dominant and active diazotrophs in the rhizospheres of various crops, including cucumber [3], grapevine [4], tobacco [5], and coffee [6], as well as in Tibetan grassland soils [7]. Furthermore, recent research identified *Skermanella* as a core member of the diazotroph community in biological soil crusts (BSCs) from temperate semi-arid and arid deserts of China [8]. These BSCs, microbial communities forming on soil surfaces in drylands, play crucial roles in soil stabilization and nutrient cycling. However, isolated *Skermanella* strains have not demonstrated nitrogen-fixing activity in traditional acetylene-based reduction assays [9,10], contrasting with evidence of their activity observed in laboratory and environmental settings. In addition, metagenomic analyses of human skin revealed the presence of *Skermanella* strains, though their ecological roles remain poorly described [11]. To date, detailed studies of the physiology and adaptive traits encoded by *Skermanella* genes in response to environmental conditions are still lacking.

Additionally, six other species of *Skermanella*, isolated as type strains from the environments, have been reported: *S. aerolata* 5416T-32^T^, from airborne particles [12]; *S. stibiiresistens* SB22^T^, from soil surrounding coal mines [10]; *S. rubra* YIM 93097^T^, from desert soil [10]; *S. rosea* M1^T^ and *S. mucosa* 8-14-6^T^, both from hydrocarbon-contaminated desert soils [13,14]; and *S. pratensis* W17^T^, from meadow soil [15]. Whole genome sequences in contig form have been reported in the NCBI database, for the strains *S. aerolata* 5416T-32^T^, *S. stibiiresistens* SB22^T^, and *S. pratensis* W17^T^. Members of the genus *Skermanella* are characterized as Gram-negative, motile, facultatively anaerobic or strictly aerobic and possessing a high DNA G+C (%) content.

In this study, a novel species belonging to the genus *Skermanella,* designated as strain TT6^T^, was isolated from human skin for its ability to degrade *t*-octylphenol polyethoxylates (Triton X-100), a compound widely used as a non-ionic detergent and wetting agent in various industrial and household products [16]. In addition, the genomes of the isolate TT6ᵀ and two type strains, *S. rosea* M1^T^ and *S. mucosa* 8-14-6^T^, were sequenced and analyzed to identify the specific genes involved in environmental adaptations. Genomic analysis revealed that these strains harbor genes implicated in CO_2_ fixation and an anaerobic photosystem II complex, with the strains TT6^T^ and *S. rosea* M1^T^ also possessing genes encoding nitrogen fixation pathways. Furthermore, we demonstrated that these latter strains exhibited growth in a nitrogen-free medium under microaerobic conditions.

## 2. Materials and Methods

### 2.1. Bacterial Strains and 16S rRNA Gene Sequencing

To isolate bacteria capable of degrading Triton X-100 from human skin, sterile cotton swabs that had been moistened with sterile deionized water were used to swab the forehead skin of male volunteers in their twenties. The swabs were directly streaked onto a minimal salt basal (MSB) agar medium, containing 0.5% (*v*/*v*) Triton X-100 (Sigma-Aldrich, Seoul, Republic of Korea). The plates were incubated at 28 °C for 1 to 3 weeks under aerobic conditions. The MSB medium contained 4% (*v*/*v*) solution A (Sol A), 1% (*v*/*v*) solution B (Sol B), and 0.5% (*v*/*v*) solution C (Sol C) [17]. Sol A was 1.0 M sodium–potassium phosphate buffer (pH 7.2–7.3), Sol B was a formulated mineral base containing 20 g L^−1^ nitrilotriacetic acid, 28.9 g L^−1^ anhydrous MgSO_4_, 6.67 g L^−1^ CaCl_2_·2H_2_O, 18.5 mg L^−1^ (NH_4_)_6_Mo_7_O_24_·4H_2_O, 198 mg L^−1^ FeSO_4_·7H_2_O, 100 mL L^−1^ Metals 44, and Sol C was 20% (*w*/*v*) of (NH_4_)_2_SO_4_ aqueous solution. After incubation, light pink colonies were streaked out on MSB agar medium supplemented with 0.2% (*w*/*v*) yeast extract and 0.2% (*w*/*v*) sodium pyruvate to obtain pure cultures. The purified strain, designated as TT6^T^, was maintained in a stock solution, containing 0.1 g L^−1^ tryptone, 0.1 g L^−1^ yeast extract, 0.05 g L^−1^ NaCl, and 60% (*v*/*v*) glycerol, and stored at −72 °C for long-term preservation. The growth of strain TT6^T^ was evaluated on nutrient (NB) agar, Luria–Bertani agar, and trypticase soy agar, with optimal growth observed on NB agar at 28 °C for three days. *S. rosea* M1^T^ (=KEMB 2255-458^T^ = JCM 31276^T^) and *S. mucosa* 8-14-6^T^ (=KEMB 2255-438^T^ = JCM 31590^T^) were obtained from the KEMB (Korea Environmental Microorganisms Bank, Suwon, Republic of Korea). *S. pratensis* W17^T^ (=KCTC 62434^T^) was obtained from the KCTC (Korean Collection for Type Cultures, Jeongeup, Republic of Korea). To identify the isolate TT6^T^, a 16S rRNA gene was amplified using the universal primer sets 27F (5′-AGAGTTTGATCCTGGCTCAG-3′) and 1492R (5′-GGTACCTTGTTACGACTT-3′) [18]. The resulting amplicons were purified using a PCR purification kit (NucleoGen, Yongin, Republic of Korea) following the manufacturer’s protocol, and the nucleotide sequence was determined at GenoTech (Daejeon, Republic of Korea) via Sanger sequencing. The 16S rRNA gene sequence (1112 bp) for strain TT6^T^ was deposited in GenBank under accession number MW314784, and was used for BLASTN searches in the NCBI database of bacterial type strains [19].

### 2.2. Genome Sequencing, Assembly, and Annotation

The total DNA from strains TT6^T^, *S. rosea* M1^T^, and *S. mucosa* 8-14-6^T^ was extracted and purified using the phenol–chloroform extraction and ethanol precipitation method. DNA quality and quantity were assessed using UV absorbance scanning (Nanodrop spectrophotometer, Scinco Co., Seoul, Republic of Korea) and fluorescence measurements (Qubit 4 fluorometer, Thermo Fisher, Waltham, MA, USA) [20]. The whole genome of strain TT6^T^ was sequenced using the PacBio RSII sequencing platform with 20 kb libraries and the Illumina NovaSeq6000 sequencing platform, with 151 paired-end reads of PCR-free 550 bp libraries (TruSeq DNA PCR-free), at DNAlink Co., Seoul, Republic of Korea. The genomes of *S. rosea* M1^T^ and *S. mucosa* 8-14-6^T^ were also sequenced, using both short-read NovaSeq6000 technology at DNAlink and the long-read MinION platform (Oxford Nanopore Technology, Oxford, UK). For Nanopore sequencing, libraries were prepared using the SQK-LSK109 kit and multiplexed using the EXP-NBD104 barcoding kit, following the manufacturer’s protocols. Sequencing was performed on a MinION sequencer (v20.10.3) using an R9.4.1-flow cell [20]. Reads were base-called and demultiplexed using Guppy v4.2.2 in high accuracy mode, and filtered for quality (minimum Q-score = 7) before conversion to FASTQ format. The sequencing platforms yielded 7.76 Gb, 7.05 Gb, and 6.27 Gb of clean data from strains TT6^T^, *S. rosea* M1^T^, and *S. mucosa* 8-14-6^T^, respectively. Long and short reads were de novo assembled using Unicycler v0.4.9b [21], and the quality of the assembled genome sequences was evaluated with CheckM v1.1.3 software [22]. The raw sequencing data and assembled genome sequences were deposited in GenBank, and annotation of the genome sequences was performed using the NCBI Prokaryotic Genome Annotation Pipeline (PGAP, www.ncbi.nlm.nih.gov/genome/annotation_prok/) [23], applying the best-placed reference protein set, GeneMarkS-2+ (ver 5.1). These depositions and anotations were processed between 4 January 2021 and 9 November 2021.

### 2.3. Genome Analyses

Whole-genome similarity metrics for strain TT6^T^ and closely related type strains were assessed using average nucleotide identity (ANI) values. Initial searches were conducted against the GenomesDB reference database, using the pairwise genome comparison service of JSpeciesWS (http://jspecies.ribohost.com/jspeciesws/) [24], accessed 9 November 2021 and last accessed on 22 December 2024. Further ANI evaluations were performed using the ANI calculators from Ezbiocloud (orthoANIu) (https://www.ezbiocloud.net/tools/ani) [25] and Kostas Lab (http://enve-omics.ce.gatech.edu/ani/). The latter two tools were last accessed on 22 December 2024. The average and standard deviation (SD) of the ANI values from these three sources were calculated. Digital DNA–DNA hybridization (dDDH) values were calculated using the Genome-to-Genome Distance Calculator (GGDC 3.0), with formula 2 (https://ggdc.dsmz.de/ggdc.php) [26], last accessed on 22 December 2024. The average amino acid identity (AAI) values for genome pair comparison were determined using the Newman Lab AAI and BBH Calculator (https://newman.lycoming.edu/AAI.html), accessed on 20 February 2023. A Venn diagram of orthologous gene clusters among *Skermanella* strains was generated using OrthoVenn2 (https://orthovenn2.bioinfotoolkits.net/home) [27], accessed on 20 February 2023. Metabolic pathways encoded in the genomes were analyzed using BlastKOALA (KEGG Orthology and Links Annotation, v2.2) (https://www.kegg.jp/blastkoala/) [28] and the Rapid Annotations using Subsystems Technology (RAST) SEED Viewer (https://rast.nmpdr.org/seedviewer.cgi) [29], both accessed on 20 July 2024. CRISPR arrays were also identified using RAST.

### 2.4. Phylogenomic Analysis

A phylogenomic tree was constructed based on single-copy genes from strain TT6^T^ and 20 type strains with rRNA gene sequences showing over 90% similarity to that of strain TT6^T^. The analysis was performed using RAxML (Randomized Accelerated Maximum Likelihood, v8.2.11) with the codon tree method, utilizing the phylogenetic tree-building service provided by the Pathosystems Resource Integration Center (PATRIC; version 3.6.11, https://www.patricbrc.org) [30], accessed on 20 February 2023 and last accessed on 10 December 2024.

### 2.5. Nitrogen Fixation and Dentification Assessments

The nitrogen-fixing ability of *Skermanella* strains was evaluated by incubating cultures at 30 °C for 14 days. The medium (40 mL) consisted of a 1% solution of SolB of MSB, 15 mM sodium–potassium phosphate buffer (pH 7.2–7.3), 0.1% (*w*/*v*) soft agar, and 80 µL of a vitamin mixture (RPMI 1640 vitamin solution, Sigma-Aldrich), contained in a test tube measuring 30 cm in height and 2.7 cm in diameter, with the medium reaching a height of 8 cm in the test tube. When necessary, 0.025% d-mannitol and/or 0.05% NH_4_Cl were added. For the denitrification assay, 0.025% d-mannitol and 0.05% NaNO_3_ were added to the medium along with other nutrients, under the same culture conditions used for the nitrogen fixation assay. The seeds of TT6^T^ and *S. rosea* M1^T^ were grown in R2A liquid medium containing 0.2% sodium pyruvate and 0.25% NaCl, while *S. mucosa* 8-14-6^T^ was grown without 0.25% NaCl. Cultures were shaken at 140 rpm at 32 °C for 24 h. The cultured seeds were then centrifuged, washed twice with saline, and inoculated after autoclave once the agar had melted.

### 2.6. Phenotypic, Biochemical, and Cellular Fatty Acid Analyses

Gram staining was performed using cells from exponentially growing cultures in an NB medium [31], and motility was observed using phase contrast microscopy (Nikon, Tokyo, Japan). Cellular morphology and flagella were observed using transmission electron microscopy (TEM) (Libra 120, Carl Zeiss, Oberkochen, Germany) with phosphotungstic acid as a negative stain. Cell motility was also tested by stab inoculating cells into plates containing an NB medium with 0.3% (*w*/*v*) agar, which were incubated at 28 °C for three days. Anaerobic respiration was tested by growing cells on an NB medium in a closed jar with an anaeroBag (Chongqing Pang Tong Medical Devices Co., Chongqing, China) for 10 days, at 28 °C. The assimilation of polyethylene glycol (PEG) 200 and PEG 1000 was tested on MSB agar at a concentration of 0.1% (*v*/*v*), where they served as the sole carbon and energy source. Biochemical and physiological analyses of strain TT6^T^ were conducted using cells grown on NB or the indicated medium for 3 days at 28 °C. All tests followed protocols provided by the American Society of Microbiology (https://asm.org/Browse-By-Content-Type/Protocols), accessed on 5 March 2022 or commonly used techniques [31,32]. Experiments were conducted in triplicate. Catalase and oxidase activities were tested by placing a 10 μL droplet of 3% (*v*/*v*) H_2_O_2_ solution and 1% (*w*/*v*) *N,N,N*′*,N*′-tetramethyl-*p*-phenylenediamine dihydrochloride (Sigma-Aldrich) onto NB agar plates and paper discs containing bacterial colonies, respectively. Bubble formation and a color change to purple within 5 min were regarded as positive indicators for each activity. DNase activity was tested by growing the cells on DNase agar with methyl green (Kisan Bio, Seoul, Republic of Korea). Coagulase activity was tested using rabbit plasma fibrinogen (Kisan Bio), followed by incubation at 35 °C for 4 h to examine clot formation. Hemolytic activity was tested on blood agar media with sheep blood, which was defibrinated (Kisan Bio) at 35 °C for 20 h. Urease activity was determined using Christensen’s urea agar [31]. β-Galactosidase activity was determined in cells grown on NB for 3 days using *o*-nitrophenyl-D-galactopyranoside (ONPG) as a substrate [32]. Hydrolysis of carboxymethyl cellulose (1.0%, *w*/*v*), starch (0.2%, *w*/*v*), skimmed milk (1.4%, *w*/*v*), Tween 20 (1.0%, *w*/*v*), and Tween 80 (1.0%, *w*/*v*) was assessed on an NB agar medium. To test the metabolism of various carbon sources, strain TT6^T^ was analyzed using the API 20 NE kit, following the manufacturer’s instructions (bioMérieux, Marcy-l’Étoile, France). To determine cellular fatty acid profiles, strain TT6^T^ was cultured at 28 °C for 48 h on an NB agar medium until the late-exponential phase. Fatty acid methyl esters were prepared according to the method described in the Sherlock Microbial Identification System (MIDI). Fatty acids were analyzed using gas chromatography (GC) (Agilent 7890 GC-FID system, Santa Clara, CA, USA), and identified utilizing the Microbial Identification software package (v. 6.3), based on the Sherlock Aerobic Bacterial Database (TSBA60) [33]. Isoprenoid quinones of strain TT6^T^ were extracted, purified, and analyzed using a reversed-phase HPLC system with a Fortis C18 column (5 µm, 4.6 mm × 150 mm), as previously described [34]. Analyses of the composition of cellular fatty acids and quinones were carried out at AceEMzyme Co. (Ansung, Republic of Korea).

## 3. Results and Discussion

### 3.1. Genome Sequencing and Phylogenomic Analyses

#### 3.1.1. Identification of Strain TT6^T^ and Genome Sequencing of TT6^T^ and Skermanella Type Strains

BLASTN searches in the NCBI type species database, using the 16S rRNA gene sequence (GenBank accession No. MW314784), confirmed that strain TT6^T^ is a member of the genus *Skermanella*. The most closely related type strains were *S. rosea* M1^T^ (99.8% similarity), *S. mucosa* 8-14-6^T^ (99.0%), and *S. pratensis* W17^T^ (98.5%). A neighbor-joining tree based on 16S rRNA gene sequences showed that these four strains formed a phylogenetic group which was distantly related to other *Skermanella* type strains (Appendix A). For genome-level analyses, the genome sequencing of strains TT6^T^, *S. rosea* M1^T^ and *S. mucosa* 8-14-6^T^ was performed using hybrid DNA sequencing platforms, as described in the Materials and Methods.

Whole genome sequencing assembly of strains TT6^T^, *S. rosea* M1^T^, and *S. mucosa* 8-14-6^T^, with an average genome coverage of 2102-, 917-, and 820-fold, respectively, yielded circular forms of contigs for the chromosomes and plasmids from each strain. CheckM analysis showed 100% completeness for all three genomes, with no strain heterogeneity. The genome sizes of TT6^T^, *S. rosea* M1^T^ and *S. mucosa* 8-14-6^T^ were 7.57 Mbp, 7.76 Mbp, and 7.84 Mbp, respectively (Table 1).

All three strains possessed 4–6 circular plasmids and a circular chromosome. A total of 6830–7069 CDS, with 17.9~18.5% hypothetical proteins and seven rRNA (5S/16S/23S) operons, were identified. In all three strains, six rRNA operons were located in the chromosome, and one in the plasmid. The G+C contents ranged between 67.2 and 67.6 mol%. Some plasmids exhibited lower G+C contents than the average for the strain, indicating their acquisition from external environments. A total of 60 to 61 tRNA, 3 ncRNA and 2 tmRNA were identified. The ncRNAs included *ssrS* (6S RNA), *rnpB* (RAase_P_RNA), and *ffs* (SRP_RNA), while the tmRNA consisted of two copies of *ssrA* (transfer-messenger RNA). The features of chromosomes and plasmids are available in Table 1. In contrast, the genome of *S. pratensis* W17^T^ is smaller (5.87 Mbp) and contains six rRNA operons (5S/16S/23S) (GenBank accession number of CP030365) [15]. In addition, CRISPR arrays with different repeat sequences were identified when comparing the *Skermanella* strains, indicating differences in bacteriophage immunity.

#### 3.1.2. Phylogenomic Analyses and Comparative Genomics

Whole genome-based phylogenomic analysis, based on alignments of 431 coding gene sequences from 20 closely related type strains, revealed a phylogenetic pattern that was similar to that observed in the 16S rRNA gene-based tree (Figure 1). Strain TT6^T^ clustered with *S. rosea* M1^T^, and separately with *S. mucosa* 8-14-6^T^ and *S. pratensis* W17^T^, forming a distinct group on the same branch. The distance between the latter two species is similar to that between the species in the first cluster. These four strains, sharing a close lineage, were distantly related to *S. stibiiresistens* SB22^T^ and *S. aerolata* 5416T-32^T^. The genera *Skermanella* and *Arenibaculum* form a distinct clade that clusters with members of the genus *Azospirillum*, which are known for their nitrogen-fixing capabilities. This clade is further related to another clade comprising genera such as *Nitrospirillum* and *Niveispirillum*, all of which belong to the family *Azospirillaceae*. The positions of the genera *Deserticella* and *Desertibacter*, which have been proposed as closely related to the genus *Skermanella* [28], could not be determined in the phylogenomic tree, due the lack of available genome information for these genera.

The genome information obtained in this study facilitated a comparative analysis of the type strains *S. rosea* M1^T^, *S. mucosa* 8-14-6^T^ and *S. pratensis* W17^T^ with strain TT6^T^ using overall genome relatedness indexes (OGRIs). Previously, the type species *S. rosea* M1^T^ and *S. mucosa* 8-14-6^T^ were validated based on the 16S rRNA gene sequence homology, phenotypic characteristics, and chemotaxonomic features [13,14]. The OrthoANIu and dDDH values between TT6^T^ and its closest type strain, *S. rosea* M1^T^, were 94.5% (56.8% average aligned genome coverage) from ANI and 64.7%, respectively. The average (SD) of ANI from three different calculations, as described in Materials and Methods, were 94.14% (±0.5%) between TT6^T^ and *S. rosea* M1^T^. The OrthoANIu, dDDH and AAI similarity values between strain TT6^T^ and related type strains are summarized in Figure 2A. These OGRIs are below the currently recognized species delineation thresholds, which are 95–96% for ANI and 70% for dDDH [29,30]. These findings suggest that TT6^T^ represents a putative novel species. The ANI and dDDH values among the four strains (TT6^T^, *S. rosea* M1^T^, *S. mucosa* 8-14-6^T^ and *S. pratensis* W17^T^) ranged from 94.53 to 91.09% and from 64.7 to 42.9%, respectively (Figure 2A). The values between each of these four strains and the next closest strain, *S. stibiiresistens* SB22^T^, were 81.43–82.39% for ANI and 24.8–25.1% for dDDH. The AAI values among the four strains were calculated to range from 94.84% to 92.55%, closely aligning with the ANI similarities, although slightly higher in each comparison. Notably, the AAI value between TT6^T^ and *S. rosea* M1^T^ (94.78%) was marginally lower than that between *S. rosea* M1^T^ and *S. mucosa* 8-14-6^T^ (94.84%). While AAI values provide reliable measures of relatedness based on shared genes between strains, there are currently no universally accepted AAI thresholds species delineation [31]. Comparative genomic analysis of the four *Skermanella* type strains using OrthoVen2 identified 4,395 shared orthologous gene clusters, representing approximately 74.6% of TT6^T^’s 5890 gene clusters (Figure 2B). The number of the unique gene clusters and singletons specific to each strain were 790, 741, 794, and 314 for TT6^T^, *S. rosea* M1^T^, *S. mucosa* 8-14-6^T^, and *S. pratensis* W17^T^, respectively. These strain-specific genes may contribute to their ecological diversification.

To assess physical genome rearrangements, the genome synteny among *Skermanella* type strains and strain TT6^T^ was analyzed using MUMmer, as provided by the RAST server. Figure 3 illustrates the relationships between the chromosomes. The analysis revealed extensive interchromosomal rearrangements, including genomic inversions and the random distributions of multiple genes across the three *Skermanella* genome structures. These synteny relationships suggest that the three strains are distantly related, despite their relative proximity in the phylogenomic tree.

### 3.2. Functional Genomic Analyses

#### 3.2.1. Genomic Analysis on Carbon Fixation and Photosynthesis

The genomic features of strain TT6^T^ and the phylogenetically related type strains were analyzed with respect to their roles in the global carbon and nitrogen cycles. Genome analyses using BlastKOALA (KEGG) and RAST indicated that strain TT6^T^ harbors all the genes required for the Calvin (reductive pentose phosphate) cycle, including those encoding ribulose-1,5-bisphosphate carboxylase (RuBisCO), a key enzyme involved in carbon fixation [35]. The genes for the large (488 aa.; IGS68_23890) and small (138 aa.; IGS68_23895) subunits of RuBisCO (*cbbLS*) are organized within a gene cluster spanning from IGS68_23885 to IGS68_23925. This cluster encodes *cbbR* (LysR transcriptional regulator)-*cbbL-cbbS-cbbE1* (ribulose-phosphate 3-epimerase)-*cbbX* (ATP-binding protein)-*cbbF* (D-fructose-1,6-bisphosphatase)-*cbbP* (phosphoribulokinase)-*cbbT1* (transketolase)-*cbbA* (fructose-bisphosphate aldolase). Similar gene clusters are also identified in *S. rosea* M1^T^ and *S. mucosa* 8-14-6^T^, indicating autotrophic adaptations.

The genomes of strain TT6^T^, *S. rosea* M1^T^ and *S. mucosa* 8-14-6^T^ contain genes related to the purple bacteria-type anoxygenic photosystem II. In strain TT6^T^, a gene cluster spanning locus tags IGS68_17035 to IGS68_17100 includes genes encoding photosynthetic reaction center proteins (*pufBA* and *pufLMC*) and components of a light harvesting complex (*pufABC* and *pufE*) [36]. Adjacent to this gene cluster, genes involved in the biosynthesis of bacteriochlorophylls (*bch* genes) and carotenoid metabolism (*crt* genes) were also identified (Figure 4). The deduced functions of the gene products within the gene cluster are summarized in Appendix A. Moreover, the genome of strain TT6^T^ includes genes encoding enzymes required for each step of the established bacteriochlorophyll biosynthesis pathway, as illustrated in Figure 5. Additional genetic loci involved in bacteriochlorophyll biosynthesis from l-glycine are listed in Appendix A. These findings suggest that *Skermanella* strains may possess phototrophic capabilities; however, no studies to date have documented a phototrophic lifestyle in this genus. Despite testing various culture conditions, we were unable to observe photosynthesis-based growth or color development under light, either in aerobic or anaerobic environments. A recent study has shown that certain species within the family *Acetobacteraceae*, which is phylogenomically related to the family *Azospirillaceae* (encompassing the genus *Skermanella*), retain photosynthetic traits [2,37]. Both families belong to the order *Rhodospirillales* and are likely to share an evolutionary lineage derived from photosynthetic bacteria.

#### 3.2.2. Genomic Analysis of Adaptations to Dry Conditions

Given that strains TT6^T^, *S. rosea* M1^T^, and *S. mucosa* 8-14-6^T^ were isolated from desert or skin environments, their genomes were analyzed for the synthesis of molecules favorable for adaptation to arid niches using KEGG pathway tools. The genomes of three strains contained a gene encoding cellulose synthase (UDP-forming) and associated genes capable of synthesizing cellulose from UDP-glucose. Additionally, a series of genes involved in starch synthesis from d-glucose 1-phosphate were identified. Cellulose is known to function as a structural scaffold in biofilm formation [38], while starch acts as a polymer that can absorb water and store carbohydrates within cells. Furthermore, genes encoding enzymes that catalyze the degradation of cellulose into glucose were identified.

In addition, a series of genes responsible for producing trehalose from UDP-glucose or starch were also identified in the genomes. Trehalose, composed of two glucose molecules, is known for its ability to prevent freezing and dehydration of cells due to its high water retention properties [39]. The genomes of these strains also contained genes involved in the biosynthesis of red carotenoids, such as spirilloxanthin and tetrahydrospirilloxanthin, which are synthesized through the unusual spirilloxanthin pathway involving lycopene and rhodopin intermediates [40]. Additionally, genes for the biosynthesis of yellow carotenoids, including 3,4-dihydrospheroidene, spheroidene, and hydroxyspheroidene, were identified as being synthesized via the spheroidene pathway through neurosporene [41]. Carotenoids play a protective role by shielding cells from radicals, oxidants, and UV radiation, and they also serve as accessory pigments in light-harvesting complexes to capture solar energy [42].

Strain TT6^T^ exhibited significant changes in colony morphology and physical properties depending on the salt concentration in the R2A agar medium. When cultured at 30 °C on a medium without added salt, the colonies displayed a liquid-like consistency. However, in the presence of 1% NaCl, the colonies became smaller, strongly adhered to the agar surface, and developed an orange color while losing mucosity (Appendix A). Similarly, in liquid culture under these salt conditions, strain TT6^T^ formed clumps in the absence of salt and demonstrated suspended growth in the presence of 1% NaCl.

#### 3.2.3. Genomic Analysis of Dentification and Nitrogen Fixation, and their Activities

Genome-based in silico analyses revealed that the strains *S. rosea* M1^T^ and *S. mucosa* 8-14-6^T^, but not strain TT6^T^, possess the genes required to catalyze the denitrification of nitrate (NO₃^−^) to dinitrogen (N_2_). This process proceeds through a series of sequential reactions involving the intermediate nitrite (NO_2_^−^), nitric oxide (NO), and nitrous oxide (N_2_O) formation, catalyzed by the enzymes nitrate reductase (encoded by *narGHI* and *napAB*), nitrite reductase (*nirS*), nitric oxide reductase (*norBC*), and nitrous oxide reductase (*nosZ*), respectively (Figure 6) [43]. Notably, these strains encode two distinct types of dissimilatory nitrate reductase: the cytoplasmic membrane-associated NarGHI, and the periplasmic NapAB [44]. This genomic analysis suggests that *S. rosea* M1^T^ and *S. mucosa* 8-14-6^T^ have the potential for anaerobic respiration utilizing nitrate as an electron acceptor. The genes involved in these sequential denitrification reactions are illustrated in Figure 6. When denitrification activity was tested using the API 20 NE system, *S. rosea* M1^T^ and *S. mucosa* 8-14-6^T^ tested positive, whereas strain TT6^T^ tested negative. Further examination of denitrification activity was conducted in test tubes as described in the Materials and Methods. Strain TT6^T^ exhibited growth restricted to the upper 0.9 mm of the medium. In contrast, *S. rosea* M1^T^ showed growth extending below this depth, accompanied by bubble formation (Figure 7A). These observations indicate that strains TT6^T^ and *S. rosea* M1^T^ preferentially utilize O_2_ and NO₃^−^ as electron acceptors under the tested conditions. Moreover, putative genes (*nxrAB*) encoding enzymes responsible for the oxidation of nitrite to nitrate, which are typically associated with chemolithoautotrophic nitrite-oxidizing bacteria [45], were identified in *S. rosea* M1^T^ not in strain TT6^T^ (Figure 6).

Notably, the genomes of TT6^T^ and *S. rosea* M1^T^ were found to contain genes (*nifDKH*) encoding Mo-nitrogenase [46], an enzyme responsible for catalyzing the conversion of N_2_ to NH_3_, known as nitrogen fixation. In contrast, these genes were absent in the genome of *S. mucosa* 8-14-6^T^. The nitrogenase-encoded genes were also identified in *S. pratensis* W17^T^, *S. aerolata* 5416T-32^T^, and *S. stibiiresistens* SB22^T^ [47]. In strain TT6^T^, a gene cluster spanning approximately 25.5 kb (from IGS68_09220 to IGS68_09355) was identified, encoding nitrogenase, nitrogenase assembly, and accessory proteins (Figure 8). The deduced gene products corresponding to each gene locus within the gene cluster are summarized in Appendix A. The *Skermanella* nitrogenase genes share the highest sequence homology with those in genus *Azospirillum*, a genus known for nitrogen-fixing and its role in promoting plant growth.

Nitrogen fixation has not yet been experimentally demonstrated in the genus *Skermanella*. Since bacterial nitrogen fixation is highly sensitive to O_2_ concentrations, the degree of sensitivity may vary among bacterial species. To account for this variability, the present experiment was conducted in test tubes containing 0.1% agar. Following autoclaving, O_2_ from the air diffuses from the surface, creating an oxygen gradient within the medium. The results showed that, in a medium containing d-mannitol as a carbon source and NH_4_Cl as a nitrogen source, strain TT6^T^ exhibited dense growth in the submerged top part of the medium, indicating aerobic metabolism (Figure 7B(a)). However, in the absence of NH_4_Cl, a microbial growth band was observed at a depth of 1.5–1.8 cm below the surface (Figure 7B(b)). No growth of TT6^T^ was observed in the control medium, which lacked d-mannitol (Figure 7B(c)). A similar growth pattern was observed in *S. rosea* M1^T^ (Figure 7B(d,e)). In contrast, *S. mucosa* 8-14-6^T^, which lacks these genes, exhibited no growth in the absence of NH_4_Cl, although it showed growth in a medium containing both d-mannitol and NH_4_Cl (Figure 7B(f,g)). When the test tubes were incubated for an extended period, distinct new growth bands formed deeper within the medium under the conditions shown in Figure 7B(b), suggesting that strain TT6^T^ cells can detect and respond to critical oxygen concentrations. These findings indicate the possibility that strains TT6^T^ and *S. rosea* M1^T^ are capable of N_2_ fixation under highly restrictive environmental conditions. The conventional nitrogen fixation assay, previously used for *Skermanella* strains, such as *S. aerolata* 5416T-32^T^ and *S. pratensis* W17^T^, involved replacing a portion of the air in the headspace (e.g., 1/3, *v*/*v*) with acetylene gas. However, this approach yielded negative results, possibly due to experimental conditions that failed to provide the critical oxygen concentrations required for nitrogen fixation by these bacteria [9,10]. For context, *Azotobacter vinelandii* is known to have three types of nitrogenases: Mo-, V-, and Fe-nitrogenases, with their catalytic components and reductase encoded by *nifDKH*, *vnfDGKH*, and *anfDGKH*, respectively. AnfG regulates the activity of Fe-nitrogenase and is reported to function independently of interactions with Mo-nitrogenase [46], which is the type present in *Skermanella*. Based on these findings, the nitrogenase activity observed in this study is presumed to be independent of AnfG activity.

Plant growth-promoting rhizobacteria (PGPR) are known to enhance plant growth by various mechanisms, including nitrogen fixation, phosphate solubilization, and phytohormone production [48]. In nitrogen-limited soils, the application of PGPR has been shown to improve plant nitrogen uptake and overall growth [49]. Although *Skermanella* species have not traditionally been recognized as nitrogen-fixing bacteria, we demonstrate their ability to grow in nitrogen-free medium under critical oxygen concentrations. Their potential plant growth-promoting traits warrant further investigation.

#### 3.2.4. Genomic Analysis of Hydrogen, Sulfur, and Triton X-100 Metabolism

All three strains possess a gene cluster encoding a nickel-dependent hydrogenase (HupSL), essential for photoautotrophic growth using H_2_ and CO_2_ as substrates. In strain TT6^T^, this cluster spans from IGS68_07430 to IGS68_07505, and is similar to the hydrogenase system of *Rhodopseudomonas palustris* (Appendix A) [50]. In *Rhodobacter capsulatus*, the HupSL hydrogenase has been demonstrated to be essential for photoautotrophic growth using H_2_ and CO_2_ as substrates [51]. In diazotrophs, the HupSL hydrogenase functions as an uptake enzyme, utilizing H_2_ generated as a by-product of nitrogen fixation [52].

The three bacterial strains possess a SOX (sulfur oxidation) system, which enables them to oxidize reduced sulfur compounds, such as thiosulfate (S_2_O_3_^2−^), to sulfate (SO_4_^2−^), thereby generating reducing equivalents for energy production. This system is primarily utilized by chemolithotrophic bacteria [53]. This reaction occurs in the periplasmic space of gram-negative bacteria through the coordinated actions of multiple enzymes. Specifically, SoxYZ acts as a carrier by binding sulfur substrates; SoxXA functions as cytochromes involved in electron transfer; SoxCD serves as sulfite dehydrogenase; and SoxB facilitates the release of sulfate through hydrolysis [53]. The Sox system (*soxYZXACDB*) in TT6^T^ is organized into two separate clusters, as detailed in Appendix A, with *soxYZ* exhibiting paralogs. Despite testing the three strains for autotrophic growth with reduced sulfur compounds, including Na_2_S, S^0^ and Na_2_S_2_O_3_, under varying light and ether aerobic or anerobic environments, as well as in 0.1% soft agar [54], no measurable growth with NaHCO_3_ or pigment development was observed after two weeks of incubation. This suggests that the Sox system may play a role in survival within specific ecological niches or might be undergoing functional loss through evolutionary processes.

Our preliminary experiment, using LC/MS analysis, revealed that strain TT6^T^ preferentially degrades higher molecular weight Triton X-100 by targeting the ethoxy chains, resulting in the accumulation of carboxylate intermediates (manuscript in preparation). It is hypothesized that dehydrogenases may play a role in this process. However, enzymes with specific dehydrogenase domain groups have not yet been identified in current studies, making it challenging to pinpoint the associated genes through genome analysis.

### 3.3. Differentiating Characteristics for Strain TT6^T^ and the Closely Related Type Species

#### 3.3.1. Differential Traits of Strain TT6^T^ Compared to Other Type Strains, and Evaluation of Strain TT6^T^ as a Novel Species

Strain TT6^T^ was further characterized, and the results are described in Section 3.3.2. The microscopic examination of strain TT6^T^ revealed the presence of a single polar or subpolar flagellum (Appendix A). The results of API kit analyses for strain TT6^T^ and closely related type species of the genus *Skermanella*, including *S. rosea* M1^T^, *S. mucosa* 8-14-6^T^, and *S. pratensis* W17^T^, are presented in Appendix A (API 20 E), Appendix A (API 20 NE), Appendix A (API 50 CH), and Appendix A (API ZYM). A summary of fatty acid composition, along with data from other type strains, is provided in Appendix A. The result of polar lipid analysis is presented in Appendix A. The major polar lipids identified in strain TT6^T^, as determined by molybdophosphoric acid spray, included phosphatidylethanolamine, phosphatidylcholine, and phosphatidylglycerol. Minor amounts of an unidentified amino phospholipid, detected by both ninhydrin and molybdenum blue staining, as well as an unidentified phospholipid and diphosphatidylglycerol, were also observed. The polar lipid composition of strain TT6^T^ is similar to that of the other three species; however, it has a lower DPG level and contains an amino phospholipid not previously found in the genus *Skermanella*. A summary of differentiating characteristics for strain TT6^T^ and its closely related type species is provided in Table 2. This study confirmed that strain TT6^T^ exhibits variations compared to its closest relative, *S. rosea* M1^T^, in physiological traits, carbon source utilization, and the composition of cellular fatty acids, polar lipids, and respiratory quinones.

In conclusion, genomic comparison data indicate that strain TT6^T^ is most closely related to *S. rosea* M1^T^, based on OGRIs. However, the ANI and dDDH values between the two strains are below the accepted thresholds for bacterial speciation, suggesting that they belong to different species. The genetic distinctness of strain TT6^T^ from its three closest phylogenetic neighbors, *S. rosea* M1^T^, *S. mucosa* 8-14-6^T^, and *S. pratensis* W17^T^, combined with variations in phenotypic and chemotaxonomic traits, supports the proposal that strain TT6^T^ represents a novel species within the genus *Skermanella*. In particular, strain TT6^T^, compared to *S. rosea* M1^T^, lacks denitrification activity, while exhibiting the ability to grow in a nitrogen-free medium, which contrasts with *S. mucosa* 8-14-6^T^. Furthermore, strains other than TT6^T^ possess genes for nitrite oxidation, which support a chemolithoautotrophic lifestyle. These differences suggest that their ecological roles in the environment may vary.

#### 3.3.2. Description of *Skermanella cutis* sp. nov.

##### *Skermanella cutis* (cu’tis. L. gen. n. *cutis*, of the Skin)

Cells are gram-stain-negative, rod shaped, measuring 2.2–4.5 μm in length and 1–1.5 μm in width, motile by a single polar or subpolar flagellum. Colonies appeared pale pink, translucent, mucoid, and circular (3–6 mm diameter) after incubation on NB agar at 28 °C for 3 days. Growth was observed at 10–40 °C (optimum, 25–30 °C), pH 5–9 (optimum, pH 6–7), NaCl concentrations of 0–3% (*w*/*v*, optimum 0.5–1.0%) on NB. Strain TT6^T^ was positive for assimilation of Triton X-100, PEG (polyethylene glycol) 200, and PEG 1000; positive for hydrolysis of starch, esculin, Tween 20, and Tween 80; positive for catalase, β-galactosidase, urease, Voges–Proskauer test, and oxidase test. Strain TT6^T^ was negative for hydrolysis of gelatin, casein, starch and cellulose, as well as for hemolysis, DNase, coagulase, indole production from l-tryptophan, arginine dihydolase. Enzyme activity tests using the API ZYM kit revealed: positive for alkaline phosphatase, esterase (C4), esterase lipase (C8), leucine arylamidase, valine arylamidase, α-glucosidase, N-acetyl-β-glucosaminidase; weakly positive for hydrolysis of esculin (β-glucosidase), acid phosphatase, cystine arylamidase, and naphthol-AS-BI-phosphohydrolase. In carbon assimilation tests using the API 20 NE kit, strain TT6^T^ assimilated d-glucose, l-arabinose, d-mannose, d-mannitol, potassium gluconate, and malic acid.

The major fatty acids identified in strain TT6^T^ (>10% of the total fatty acids) were summed feature 8 (C18:1 ω7c/C18:1 ω6c, 73.5%). Minor fatty acids (<10%, >1%) included C16:0 (6.6%), summed feature 2 (C14:0 3OH/C16:1 iso I, 5.1%), C16:0 3OH (3.3%), C16:1 ω11c (2.2%), C18:1 2OH (1.8%), C18:1 ω9c (1.8%), C17:1 ω6c (1.6%), summed feature 3 (C16:1 ω7c/C16:1 ω6c, 1.5%), and C18:0 (1.3%). The only respiratory quinone detected in strain TT6^T^ was Q-10. The major polar lipids identified were phosphatidylethanolamine, phosphatidylcholine, and phosphatidylglycerol. Additionally, minor amounts of an unidentified amino phospholipid, an unidentified phospholipid and diphosphatidylglycerol were also observed. The type strain is TT6^T^ (= KCTC 82306^T^ = JCM 34945^T^), isolated from human skin swab sample in Republic of Korea. The DNA G+C content of the type strain is 67.6 mol% (genome). The GenBank accession numbers of the 16S rRNA gene and whole genome sequences of strain TT6^T^ are MW314784 and ASM1665363.2, respectively.

## 4. Conclusions

In this study, we performed whole genome sequencing of two *Skermanella* type strains and *S. cutis* TT6^T^, isolated from human skin. The genomic information obtained provides a foundational framework for understanding not only these strains but also the broader *Skermanella* genus. Genome analyses revealed the presence of previously unrecognized genes associated with those involved in global carbon, nitrogen, and sulfur cycles. In particular, the three strains possess gene clusters associated with photosystem II-based photosynthesis, hydrogen oxidation, and the oxidation of reduced sulfur compounds, indicating their potential to generate energy to support an autotrophic lifestyle via the Calvin cycle. This study also confirmed that strain TT6^T^ belongs to a new species within the genus *Skermanella*.

From a practical perspective, *S. cutis* TT6^T^, unlike *S. rosea* M1^T^, does not perform denitrification, but demonstrates growth in oxygen-limited, nitrogen-free media. This capability of *S. cutis* TT6^T^ may be beneficial for agricultural applications in oxygen-limited environments, such as the rhizosphere. Furthermore, the frequent isolation and dominant occurrence of *Skermanella* strains in arid environments, such as deserts, suggest their potential role in improving the biological conditions of these extreme ecosystems. Future molecular and biochemical studies, focusing on drought tolerance, photosynthesis, inorganic sulfur oxidation, and inorganic nitrogen metabolism, including nitrogen fixation, are needed to further elucidate the ecological and functional roles of the *Skermanella* genus.

## Figures and Tables

**Figure 1 microorganisms-13-00094-f001:**
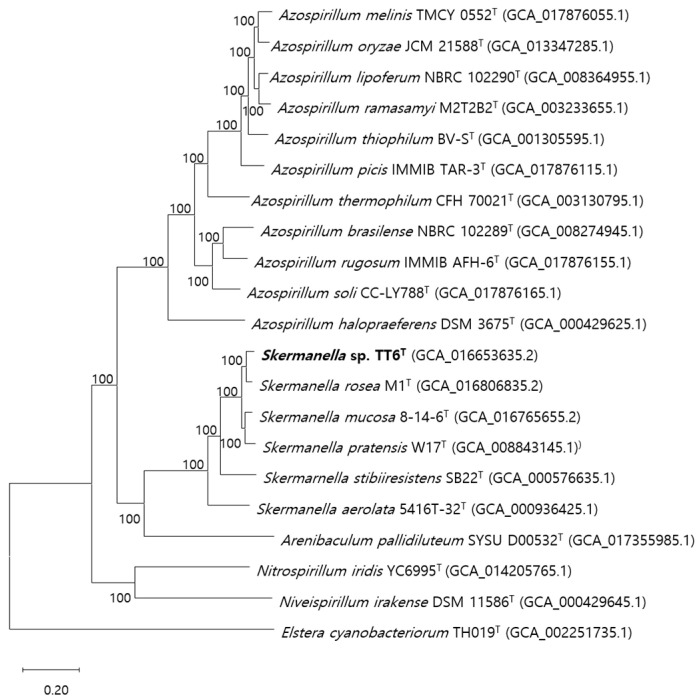
Phylogenomic tree based on a total of 431 genes of strains TT6^T^, *S. rosea* M1^T^, *S. mucosa* 8-14-6^T^, and other closely related type strains. The NCBI GenBank accession numbers for each strain are shown in brackets. The RAxML method from PATRIC was used for gene comparisons. *Elstera cyanobacteriorum* TH019^T^ was used as the outgroup. The bar indicates sequence divergence.

**Figure 2 microorganisms-13-00094-f002:**
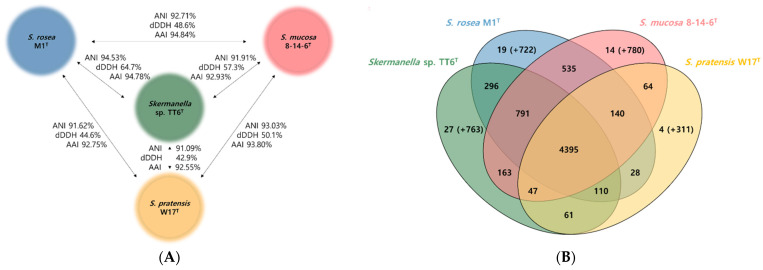
Relationship among *Skermanella* strains. (**A**) Based on orthoANIu, dDDH, and AAI values; (**B**) Venn diagram showing shared and unique orthologous gene clusters. The number of singletons specific to each genome is presented in parentheses.

**Figure 3 microorganisms-13-00094-f003:**
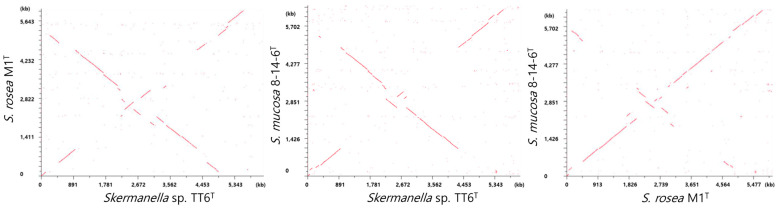
Chromosome-wise alignments of TT6^T^, *S. rosea* M1^T^, and *S. mucosa* 8-14-6^T^ against each other using MUMmer.

**Figure 4 microorganisms-13-00094-f004:**
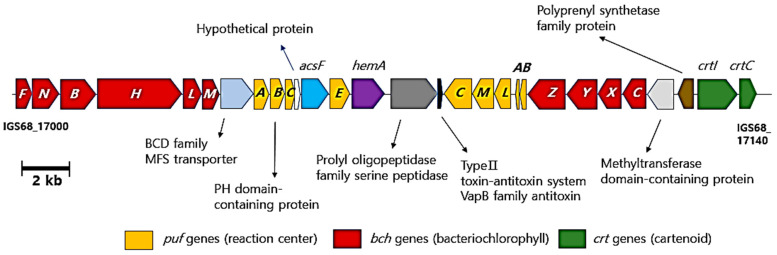
Gene cluster in strain TT6^T^-containing genes for a purple bacteria-type anoxygenic photosystem II, and the biosynthesis of light harvesting pigments. The presence of these genes indicates the potential for strain TT6^T^ to perform anoxygenic photosynthesis in low light and under anaerobic conditions, similar to that for purple photosynthetic bacteria.

**Figure 5 microorganisms-13-00094-f005:**
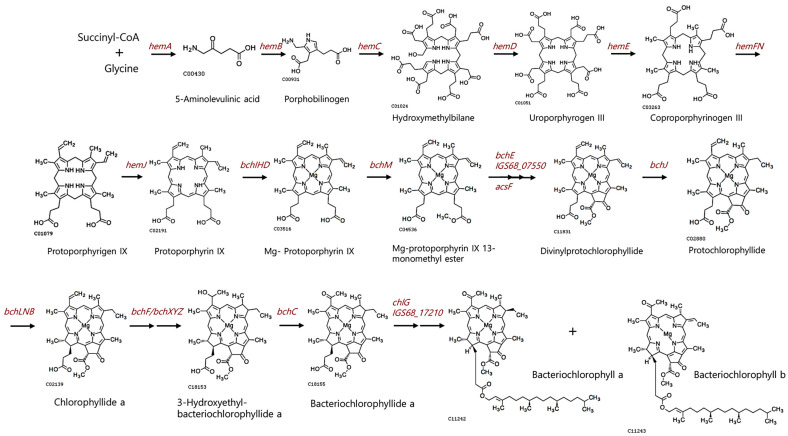
A biosynthetic pathway for bacteriochlorophylls in strain TT6^T^. This pathway is inferred from genes encoding enzymes that catalyze each step of the established bacteriochlorophyll biosynthesis pathway. The molecular structures are adapted from KEGG.

**Figure 6 microorganisms-13-00094-f006:**
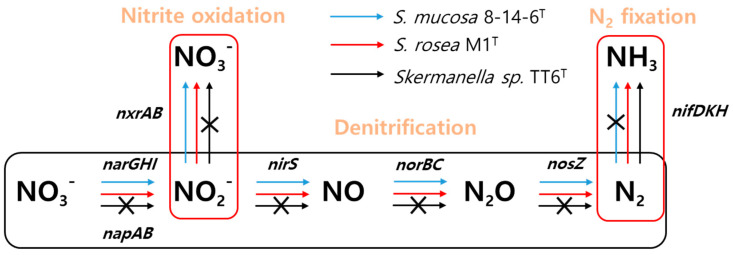
Genes and metabolic pathways involved in inorganic nitrogen metabolism in *Skermanella* strains, illustrating key processes such as denitrification, nitrite oxidation, and nitrogen fixation. Reaction steps are represented by arrows.

**Figure 7 microorganisms-13-00094-f007:**
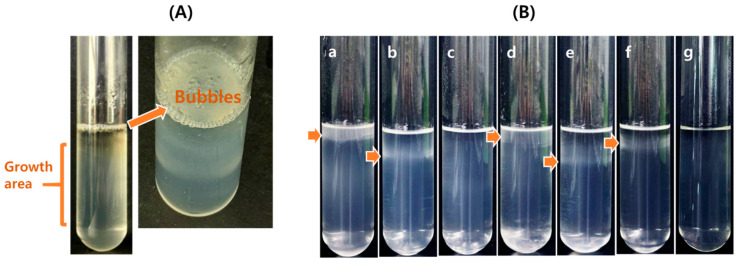
Growth of *Skermanella* strains. (**A**) Bubble formation by *S. rosea* M1^T^ under denitrification conditions. Initially, the bubbles were trapped within the medium; (**B**) cells were grown in test tubes containing 0.1% agar in the presence of d-mannitol (abbreviated as C) with and without nitrogen source of NH_4_Cl (abbreviated as N). (**a**) *Skermanella* TT16^T^, +C+N; (**b**) *Skermanella* TT16^T^, +C–N; (**c**) *Skermanella* TT16^T^, –C+N; (**d**) *S. rosea* M1^T^, +C+N; (**e**) *S. rosea* M1^T^, +C–N; (**f**) *S. mucosa* 8-14-6^T^, +C+N; (**g**) *S. mucosa* 8-14-6^T^, +C–N. Arrows indicate the positions of cell growth in the test tube.

**Figure 8 microorganisms-13-00094-f008:**
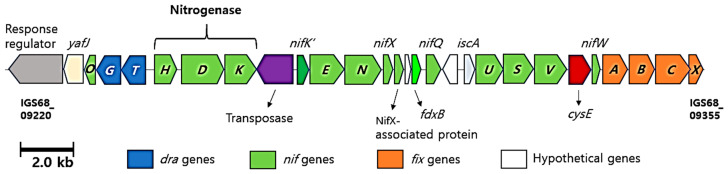
Gene cluster in strain TT6^T^ encoding nitrogenase, nitrogenase assembly, and accessory proteins.

**Table 1 microorganisms-13-00094-t001:** Overall genomic features of strain TT6^T^ and closely related *Skermanella* type strains *.

Characteristic	*Skermanella* sp. TT6^T^	*S. rosea* M1^T^	*S. mucosa* 8-14-6^T^
GenBank accession No.	CP067420~5	CP086111~7	CP086106~10
Genome size (bp)	7,569,752	7,759,404	7,840,710
Chromosome (bp)	5,940,203	6,098,631	6,407,183
No. of plasmids (total bp)	5 (1,629,549)	6 (1,660,773)	4 (1,433,527)
G+C content (mol%)	67.6	67.6	67.2
No. of total genes	7054	7292	7286
No. of protein-coding genes	6830	7069	7030
(% of hypothetical proteins)	18.5	18.4	17.9
No. of rRNAs (5S/16S/23S)	7/7/7	7/7/7	7/7/7
No. of tRNAs	61	61	60
No. other RNAs	5	5	5
No. of pseudogenes	137	136	170
No. of CRISPR arrays	5	3	2

* The general features of the genomes were analyzed using the NCBI prokaryotic genome annotation pipeline.

**Table 2 microorganisms-13-00094-t002:** Differentiating characteristics for strain TT6^T^ and closely related type species of the genus *Skermanella*. **Strains**: 1, TT6^T^; 2, *S. rosea* M1^T^; 3, *S. mucosa* 8-14-6^T^; 4, *S. pratensis* W17^T^. All data were obtained from this study unless otherwise indicated. All strains share the following characteristics: positive for hydrolysis of starch, Tween 80; positive for catalase, β-galactosidase, and oxidase activity; acid produced from l-arabinose, d-ribose, d-xylose, l-xylose, d-galactose, d-fructose, d-mannose, d-lyxose, d-fucose, l-fucose; assimilation of l-arabinose, d-xylose, d-mannose, d-fucose; positive for alkaline phosphatase, esterase (C4), esterase lipase (C8), leucine arylamidase, and valine arylamidase; and weakly positive for hydrolysis of esculin (β-glucosidase), cystine arylamidase, and naphthol-AS-BI-phosphohydrolase. Negative for lipase (C14), trypsin, α-chymotrypsin, α-galactosidase, β-glucuronidase, α-mannosidase, and α-fucosidase; negative for hydrolysis of gelatin, casein and cellulose; negative for hemolysis, DNase, coagulase, indole production from l-tryptophan, arginine dihydrolase, lysine decarboxylase, tryptophan deaminase, H_2_S production, ornithine decarboxylase, citrate utilization; and negative for assimilation of carbon sources in API 50 CH, which were excluded from the table. +, positive; w, weakly positive; −, negative.

	1	2	3	4
Source	Forehead Skin	Desert Sands ^i^	Crude Oil-Contaminated Soil ^ii^	Meadow Soil *
**Cell size (µm)**	1.0–1.5 × 2.2–4.5	(0.8–0.9 × 1.4–1.5) ^i^	(0.5–0.6 × 1.2–1.4) ^ii^	(0.7–1.2 × 1.1–1.5) *
**Colony diameter (mm)**	3–6	2–5	1–4	3–6
**Temperature range (°C)**	10–40	10–45	10–37	10–40
**NaCl range (%, *w*/*v*)**	0–3	0–2	0–1	0–5
**pH range** **N_2_ fixation genes**	5–9 +	6–9 +	7–9 −	6–9 +
**Nitrate reduction**	−	**+**	+	+
**PEG1000, PEG200 catabolism**	+	−	−	−
**Voges–Proskauer**	w	− (+) *	+	+
**Urease**	+	+ (−) ^i^	−	−
**Acid production (API 20 E, API 50 CH)**				
d-Arabinose	+	+	−	**+**
d-Glucose	+	+	−	+
l-Rhamnose	+	+	−	+
d-Mannitol	+	+	+	−
d-Cellobiose	−	−	+	−
d-Maltose	−	−	+	−
d-Melibiose	−	+	+	+
d-Saccharose (sucrose)	−	−	+	−
d-Trehalose	−	−	+	−
Amidon (starch)	−	−	+	−
Gentiobiose	−	−	+	−
d-Arabitol	+	+	+	−
**Assimilation (API 20 NE, API 50 CH)**				
Capric acid	−	−	−	+ (−) *
Adipic acid	−	w (+) *	− (w) *	w (−) *
Malic acid	w	−	−	+ (−) *
Glycerol	−	−	−	+
Erythritol	−	−	−	+
d-Arabinose	−	+	−	+
d-Ribose	−	+	+	+
l-Xylose	+	+	−	+
d-Adonitol	−	−	−	+
d-Galactose	−	+	+	+
d-Glucose	+	+	−	+
d-Fructose	+	+	−	+
l-Rhamnose	+	−	−	+
d-Mannitol	+	+	+	−
d-Cellobiose	−	+	−	−
d-Maltose	−	−	+	+
d-Lactose	+	+	+	−
d-Melibiose	−	−	+	−
d-Saccharose (sucrose)	−	−	+	−
d-Trehalose	−	−	+	+
Inulin	−	+	−	−
Amidon (starch)	−	−	+	−
Gentiobiose	+	−	−	−
d-Lyxose	−	+	−	+
l-Fucose	+	+	−	+
Potassium gluconate	+	+	−	+
Potassium 2-ketogluconate	+	−	−	+
Potassium 5-ketogluconate	−	−	−	+
**Enzyme activity (API ZYM)**				
Acid phosphatase	w	w (+) *	+	− (+) *
α-Glucosidase ^‡^	+	w (+) *	w (+) *	+
β-Glucosidase	−	w (−) *	w (+) *	w (−) *
N-Acetyl-β-glucosaminidase	+	w (−) *	w (−) *	−
**Respiratory quinones**	Q-10 (100%)	Q-10 (>90%), Q-8 (<8%), Q-6 (<2%) ^i^	Q-10 (>90%), Q-8 (<8%), Q-6 (<2%) ^ii^	Q-10 (100%) *
**Major Polar Lipids**	PE, PG, PC, DPG, PL, UL, APL	PE, PG, PC, DPG, PL, UL, AL ^i^	PE, PG, PC, DPG, PL, UL, AL ^ii^	PE, PG, PC, DPG, PL, GL *

^i^ Data from the literature [13]; ^ii^ Data from the literature [13,14]; * Data from the literature [15]. Polar lipid abbreviations: PG, phosphatidylglycerol; PE, phosphatidylethanolamine; PC, phosphatidylcholine; DPG, diphosphatidylglycerol; PL, phospholipid; APL, aminophospholipid; AL, aminolipid; GL, glycolipid; UL, unknown lipid. ^‡^ The substrate is 6-Br-2-naphthyl-β-D-galactopyranoside, which exhibits a preference for aromatic aglycones. Consequently, maltose and starch in this table could test negative for metabolism.

## Data Availability

The GenBank accession numbers for the 16S rRNA gene and complete genome sequence of strain TT6^T^ are MW314784 and CP067420~5, respectively. The GenBank accession numbers for the complete genome sequences of *Skermanella rosea* M1^T^ and *Skermanella mucosa* 8-14-6^T^ are CP086111~7 and CP086106~10, respectively.

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
