# Peer review of "Analysis of the Genomes and Adaptive Traits of Skermanella cutis sp. nov., a Human Skin Isolate, and the Type Strains Skermanella rosea and Skermanella mucosa"

_microorganisms, 2025, doi:10.3390/microorganisms13010094_

Round 1

Reviewer 1 Report

Comments and Suggestions for Authors

The manuscript microorganisms-3386424 describes the genomic, morphological and physiological-biochemical characteristics of the strain TT6 isolated from human skin. The authors describe the strain TT6 as a representative of a new species of the genus Skermanella. In my opinion, the authors need to conduct several additional studies to confirm their hypotheses.

Remarks:

1. The authors describe a new species in the manuscript, but there is no mention of this new species in either the title or the abstract of the manuscript. This is unacceptable, since the abstract should reflect all the main results of the manuscript.

2. Strain TT6 is close in genomic characteristics to the type strain of Skermanella rosea. ANI and dDDH values are lower, but very close to species boundary. However, both methods (ANI and dDDH) have an error (see, for example, https://doi.org/10.1007/s10482-017-0844-4). ANI Calculator Kostas Lab calculates the ANI value between these genomes as 94.22% (SD: 2.91%). It would be more convincing if the authors provided more results demonstrating sufficient differences between strain TT6 and the species Skermanella rosea.

3. I do not agree with the authors that they demonstrated nitrogen fixation by strain TT6 in their experiments. We can only say that the growth of the bacterial culture in a nitrogen-free medium is observed. This is not necessarily accompanied by nitrogen fixation. The method proposed by the authors has long been used to isolate azospirilla, but not all strains isolated by this method subsequently demonstrate "true" nitrogen fixation. The authors must determine the enzymatic activity of nitrogenase or demonstrate a quantitative increase in nitrogen in the strain TT6 culture. And an additional question regarding the nitrogen-free medium: 0.25 g/l mannitol - was this the only source of carbon in the medium? Nitrogen fixation is a very energy-consuming process. Is this amount of energy source sufficient for nitrogen fixation? Ashby's medium contains 20-40 times more carbon.

4. The manuscript provides descriptions of a large number of genome fragments that may be involved in various processes beneficial for strain TT6. However, the experimental part of the manuscript provides very little experimental evidence of the involvement of the described genes in the activity of strain TT6. For example, tregolose protects cells from drying out. Is this observed in strain TT6? Provide the results of the osmotic stress experiment. Conversely, strain TT6 was isolated as a potential degrader of Triton X-100. Is this strain a degrader? Which genes are involved in the degradation of Triton X-100? What end products are formed? The genome of strain TT6 contains genes of photosystem II. Have the authors investigated the ability of strain TT6 to carry out photosynthesis under anaerobic conditions in the light, using hydrogen sulfide as a source of electrons?

Author Response

The manuscript microorganisms-3386424 describes the genomic, morphological and physiological-biochemical characteristics of the strain TT6 isolated from human skin. The authors describe the strain TT6 as a representative of a new species of the genus Skermanella. In my opinion, the authors need to conduct several additional studies to confirm their hypotheses.

We appreciate the detailed review of our manuscript. The comments will greatly contribute to improving its quality.

Comments 1: The authors describe a new species in the manuscript, but there is no mention of this new species in either the title or the abstract of the manuscript. This is unacceptable, since the abstract should reflect all the main results of the manuscript.

Response 1: Thank you for pointing out this. We agree with this comment. Therefore,

  1. We have changed the title of manuscript to ‘Analyses of Genomes and Adaptive Traits of Skermanella cutis sp. nov., a Human Skin Isolate, and Type Strains Skermanella rosea and Skermanella mucosa’ (p.1, line 3). In main text and supplementary,
  2. In the text, figures, and supplementary materials, the nomenclature of strain TT6 follows the guidelines of IJSEM, and thus all instances of "TT6" have been changed to "TT6ᵀ."
  3. In abstract, revised p.1, lines 20-24, the following has been corrected. ‘Phenotypic and chemotaxonomic traits of strain TT6ᵀ were also compared with closely related type strains, confirming its genotypic and phenotypic distinctiveness. The new species, Skermanella cutis sp. nov., is proposed, with TT6ᵀ (=KCTC 82306ᵀ =JCM 34945ᵀ) as the type strain. This study underscores the agricultural and ecological significance of the genus Skermanella.’
  4. p. 13, lines, 489-500. In conclusion, genomic comparison data indicate that strain TT6ᵀ is most closely related to S. rosea M1ᵀ based on OGRIs. However, the ANI and dDDH values between the two strains are below the accepted thresholds for bacterial speciation, suggesting that they belong to different species. The genetic distinctness of strain TT6T from its three closest phylogenetic neighbors, S. rosea M1ᵀ, S. mucosa 8-14-6ᵀ, and S. pratensis W17ᵀ, combined with variations in phenotypic and chemotaxonomic traits, supports the proposal that strain TT6T represents a novel species within the genus Skermanella. In particular, strain TT6T, compared to S. rosea M1ᵀ, lacks denitrification activity, while exhibiting the ability to grow in nitrogen-free medium, which contrasts with S. mucosa 8-14-6ᵀ. Furthermore, strains other than TT6T possess genes for nitrite oxidation, which support a chemolithoautotrophic lifestyle. These differences suggest that their ecological roles in the environment may vary.

Comments 2: Strain TT6 is close in genomic characteristics to the type strain of Skermanella rosea. ANI and dDDH values are lower, but very close to species boundary. However, both methods (ANI and dDDH) have an error (see, for example, https://doi.org/10.1007/s10482-017-0844-4). ANI Calculator Kostas Lab calculates the ANI value between these genomes as 94.22% (SD: 2.91%). It would be more convincing if the authors provided more results demonstrating sufficient differences between strain TT6 and the species Skermanella rosea.

Response 2: Thank you for pointing out this. We agree with this comment.

According to OGRIs values, ANI is considered as the threshold for bacterial species differentiation at 95–96%, “or” dDDH at 70%. The dDDH value between TT6 and S. rosea M1T is 64.7%, indicating that the two strains do not belong to the same species based on the established criteria. Notably, the average aligned genome coverage for ANI is 56.8% (p. 6-7, lines 257-258), suggesting a low overall genetic homology. Furthermore, genome synteny analysis (Fig. 3) reveals significant genetic rearrangements. Additionally, in the ‘Taxonomy Check’ section of the NCBI genome database, the genus_match status is noted for strain TT6. Although TT6 is closely related to S. rosea, it is not classified as the same species (https://www.ncbi.nlm.nih.gov/datasets/genome/GCF_016653635.2/). Other physiological variations compared to S. rosea are detailed in Table 2 and discussed above. Particularly, differences in anaerobic respiration with NO3- and the potential to oxidize NO2- as a nitifier could result in differing ecological niches, underscoring the substantial distinction between these two strains as separate species.

Comments 3: I do not agree with the authors that they demonstrated nitrogen fixation by strain TT6 in their experiments. We can only say that the growth of the bacterial culture in a nitrogen-free medium is observed. This is not necessarily accompanied by nitrogen fixation. The method proposed by the authors has long been used to isolate azospirilla, but not all strains isolated by this method subsequently demonstrate "true" nitrogen fixation. The authors must determine the enzymatic activity of nitrogenase or demonstrate a quantitative increase in nitrogen in the strain TT6 culture. And an additional question regarding the nitrogen-free medium: 0.25 g/l mannitol - was this the only source of carbon in the medium? Nitrogen fixation is a very energy-consuming process. Is this amount of energy source sufficient for nitrogen fixation? Ashby's medium contains 20-40 times more carbon.

Response 3: Thank you for pointing out this. We agree fully with this comment. Therefore,

The term ‘Nitrogen fixation activity’ has been replaced with ‘the ability to grow in a nitrogen-free medium’ throughout the manuscript including abstract (p1, lines 15-16) and text (p. 11-12, lines 408-436) While the experimental results are presumed to originate from nitrogen fixation activity, directly measuring the proposed nitrogen fixation activity requires sophisticated experiments. This is because the phenomenon occurs in 0.1% agar, making cell recovery and ensuring activity during the experiment challenging. Therefore, the proposed experiments will be conducted in future studies. As suggested, in the current paper, we report results showing that both TT6 and S. rosea can grow in a nitrogen-free medium under microaerobic conditions. In addition, we tested a 10-fold higher concentration of mannitol for the experiment. The growth patterns and levels were nearly identical, indicating that N2 diffusion becomes growth-limiting with mannitol concentrations even exceeding 0.25 g/L of D-mannitol.

Comments 4: The manuscript provides descriptions of a large number of genome fragments that may be involved in various processes beneficial for strain TT6. However, the experimental part of the manuscript provides very little experimental evidence of the involvement of the described genes in the activity of strain TT6. For example,

Comments 4-1: tregolose protects cells from drying out. Is this observed in strain TT6?

Response 4-1: The purpose of this study is to perform a complete genome analysis of species belonging to the genus Skermanella and to infer genes essential for environmental adaptation, with a focus on identifying novel species. This study identified the presence of trehalose biosynthesis gene clusters, which may contribute to survival under drying conditions. However, experiments on the environmental adaptation mechanisms of these bacteria will be pursued as a separate research topic.

Comments 4-2: Provide the results of the osmotic stress experiment.

Response 4-2: In this study, it was observed and reported that the morphology and pigment production of the TT6 strain significantly changed with varying salt concentrations in the medium (supplementary Figure S2). The osmotic stress experiment, which could be part of the environmental adaptation mechanisms of these bacteria, will be pursued as a separate research topic.

Comments 4-3: Conversely, strain TT6 was isolated as a potential degrader of Triton X-100. Is this strain a degrader? Which genes are involved in the degradation of Triton X-100? What end products are formed?

Response 4-3: We have completed the proposed experiments, and LC/MS analysis revealed that strain TT6 preferentially degrades higher molecular weight Triton X-100 by targeting the ethoxy chains, leading to the accumulation of carboxylate intermediates. It is speculated that dehydrogenases may be involved in this process. However, enzymes with specific dehydrogenase domain groups have not yet been identified in current research, making it difficult to pinpoint related genes through genome analysis. We are currently preparing a manuscript on the metabolism of Triton X-100 by strain TT6 and other human skin bacteria.

Comments 4-4: The genome of strain TT6 contains genes of photosystem II. Have the authors investigated the ability of strain TT6 to carry out photosynthesis under anaerobic conditions in the light, using hydrogen sulfide as a source of electrons?

Response 4-4: We conducted multiple experiments using reduced sulfur compounds, including Na₂S, S⁰, and Na₂S₂O₃, under varying light and either aerobic or anaerobic conditions, including 0.1% soft agar. However, we were unable to observe visible growth and pigment formation. The reason for this remains unclear and has been mentioned in the revised manuscript (p. 12, lines 461-464).

Reviewer 2 Report

Comments and Suggestions for Authors

The manuscript presents an intriguing study on Skermanella sp. TT6, offering valuable insights into its genomic and physiological traits. The topic is original and contributes to the understanding of adaptive traits and potential applications of this genus. The genomic analyses are robust, with well-detailed methodologies and thoughtful comparisons with related strains. However, areas need improvement in data presentation, interpretation of results, and alignment with recent literature. Additionally, the manuscript requires better organization and more precise articulation in some sections.

1.      The genomic analyses, including ANI, dDDH, and AAI evaluations, are thorough. However, the manuscript lacks a clear explanation of how the identified pathways contribute to the ecological role of TT6.

2.      The experimental design for nitrogen fixation is novel but requires additional details regarding controls and replicates to ensure reproducibility.

3.    The authors should include the potential application of strain TT6 in agriculture, which needs to be explained with support from the literature, especially in nitrogen-limited environments.

4.      Figures and tables are generally informative but need consistent formatting. Figures from the supplementary material, such as the growth patterns in test tubes (Figure 8), should be better integrated with the main text.

5.      Figure 4 (photosystem gene clusters) should have a more detailed legend to explain the biological relevance of the illustrated pathways.

6.      Results and discussion sections are occasionally repetitive, especially when describing genomic features shared among strains.

Author Response

The manuscript presents an intriguing study on Skermanella sp. TT6, offering valuable insights into its genomic and physiological traits. The topic is original and contributes to the understanding of adaptive traits and potential applications of this genus. The genomic analyses are robust, with well-detailed methodologies and thoughtful comparisons with related strains. However, areas need improvement in data presentation, interpretation of results, and alignment with recent literature. Additionally, the manuscript requires better organization and more precise articulation in some sections.

We appreciate the detailed review of our manuscript. The comments will greatly contribute to improving its quality.

Comments 1: The genomic analyses, including ANI, dDDH, and AAI evaluations, are thorough. However, the manuscript lacks a clear explanation of how the identified pathways contribute to the ecological role of TT6.

Response 1: Thank you for pointing this out. We agree with this comment. Therefore, in section 3.2 (Functional Genomic Analyses), we have included an explanation in each subsection regarding how the identified pathways contribute to the lifestyle of strain TT6 (p.8 line 304; p.8, lines 315-316; p.9 lines 352-354; p.10, lines 379-383; p.11, lines 423-426; p.12, lines 449-451; p.12, lines 454-455). Additionally, we have incorporated a discussion of how some pathways relate to an autotrophic lifestyle in the conclusions section (p. 15, lines 557-567).

Comments 2: The experimental design for nitrogen fixation is novel but requires additional details regarding controls and replicates to ensure reproducibility.

Response 2: Thank you for pointing this out. We agree with this comment. We repeated the experiments several times and obtained consistent results. The controls are shown in Figures 7B-c and 7B-g. In Figure 7B-c, strain TT6 showed no growth when NH₄Cl was present but no carbon source was provided, indicating the growth is coupled to a carbon source. In Figure 7B-g, S. mucosa, which lacks nitrogenase genes, did not show growth without NH₄Cl, confirming no contamination of nitrogen source in the prepared medium.

Comments 3: The authors should include the potential application of strain TT6 in agriculture, which needs to be explained with support from the literature, especially in nitrogen-limited environments.

Response 3: Thank you for pointing out this. We agree with this comment. The suggestion has been included in the revised manuscript (p. 12, lines 437-443) as follows:

Plant Growth-Promoting Rhizobacteria (PGPR) are known to enhance plant growth by various mechanisms, including nitrogen fixation, phosphate solubilization, and phytohormone production [48]. In nitrogen-limited soils, the application of PGPR has been shown to improve plant nitrogen uptake and overall growth [49]. Although Skermanella species have not traditionally been recognized as nitrogen-fixing bacteria, we demonstrate their ability to grow in nitrogen-free medium under critical oxygen concentrations. Their potential plant growth-promoting traits warrant further investigation.

In conclusion section: p. 15 lines 564-569 as follows:

From a practical perspective, S. cutis TT6T, unlike S. rosea M1T, does not perform denitrification but demonstrates growth in oxygen-limited, nitrogen-free media. This capability of S. cutis TT6T may be beneficial for agricultural applications in oxygen-limited environments, such as the rhizosphere. Furthermore, the frequent isolation and dominant occurrence of Skermanella strains in arid environments, such as deserts, suggest their potential role in improving the biological conditions of these extreme ecosystems.

Comments 4: Figures and tables are generally informative but need consistent formatting. Figures from the supplementary material, such as the growth patterns in test tubes (Figure 8), should be better integrated with the main text.

Response 4: Thank you for pointing out this. We agree with this comment. Figure S3 (denitification results) has been included in the main text as Figure 7A (p. 11). Additionally, the figure numbers and corresponding text have been rearranged accordingly.

Comments 5: Figure 4 (photosystem gene clusters) should have a more detailed legend to explain the biological relevance of the illustrated pathways.

Response 5: Thank you for pointing out this. We agree with this comment. In Figure 4 legend, the following has been added (p. 8 lines 325-328). “Gene cluster in strain TT6T containing genes for a purple bacteria-type anoxygenic photosystem II and the biosynthesis of light-harvesting pigments. The presence of these genes indicates the potential of strain TT6T to perform anoxygenic photosynthesis under low-light and anaerobic conditions, similar to that of purple photosynthetic bacteria.”

Comments 6: Results and discussion sections are occasionally repetitive, especially when describing genomic features shared among strains.

Response 6: Thank you for pointing out this. We agree with this comment. Therefore, the changes have been made in the revision as follows:

  1. (original) indicating conserved pathways for carbon fixation among these strains -> (changed) indicating autotrophic adaptations (p. 8, line 304).
  2. (original) In the three strains, a gene cluster encoding a nickel-dependent hydrogenase (HupSL) was identified. This cluster also includes genes encoding proteins essential for the assembly of the nickel-dependent hydrogenase complex. In strain TT6, the corresponding gene cluster is located between IGS68_07430 and IGS68_07505 (Table S4). The gene cluster shows the highest similarity to that of Rhodopseudomonas palustris [48].

-> (changed) All three strains possess a gene cluster encoding a nickel-dependent hydrogenase (HupSL), essential for photoautotrophic growth using H₂ and CO₂ as substrates. In strain TT6T, this cluster spans from IGS68_07430 to IGS68_07505 and is similar to the hydrogenase system of Rhodopseudomonas palustris (Table S4) [50]. (p. 12, line 445-448).

  1. In the conclusion section, the details about ‘the nitrogen fixation part’ have been deleted.

Round 2

Reviewer 1 Report

Comments and Suggestions for Authors

The authors have made revisions that have improved the manuscript. However, I find that some aspects still require clarification.

1. The discussion of the phylogenetic differences between strain TT6 and S. rosea M1 should be more detailed. The authors do not indicate this, but as I understand it, the difference between strain TT6 and S. rosea M1 is the smallest among all species of the genus Skermanella. Accordingly, this level of difference between species of the genus Skermanella can become a reference for describing new species for this genus in the future. In this case, a more substantiated demonstration is required that the ANI values of 94.5% (±??) and dDDH = 64.7±?? % are sufficient to classify the strain as a new species.

In addition, the title of Figure 1 needs to be changed, since the authors use a variant of MLSA to create the tree. In this case, "Phylogenomic tree based on whole genome sequences…" is incorrect. This is not a tree based on ANI values.

2. The authors' answers regarding drought tolerance or Triton X-100 degradation should be reflected in some form in the Discussion and Conclusion sections, without hindering further work by the authors on these topics. Without such additions, the manuscript appears logically incomplete on these issues.

I also see some inconsistencies in the manuscript. The authors discuss trehalose biosynthesis genes and the possibility of strain TT6 being drought tolerant, but Tables 2 and S8 show this strain as being unable to metabolize trehalose. Also, Table S9 shows that strain TT6 is α-glucosidase positive, but Tables 2, S7, and S8 show that this strain does not metabolize maltose and starch. I find these inconsistencies should be discussed by the authors in the manuscript.

Author Response

The authors have made revisions that have improved the manuscript. However, I find that some aspects still require clarification.

We appreciate the detailed review of our manuscript. The comments will greatly contribute to improving its quality.

Comments 1-1: The discussion of the phylogenetic differences between strain TT6 and S. rosea M1 should be more detailed. The authors do not indicate this, but as I understand it, the difference between strain TT6 and S. rosea M1 is the smallest among all species of the genus Skermanella. Accordingly, this level of difference between species of the genus Skermanella can become a reference for describing new species for this genus in the future. In this case, a more substantiated demonstration is required that the ANI values of 94.5% (±??) and dDDH = 64.7±?? % are sufficient to classify the strain as a new species.

Response 1-1: Thank you for pointing out this. We agree with this comment. Our explanation is as follows:

Classification based on bacterial whole-genome sequences follows the "or" criterion for ANI (95-96%) and dDDH (70%) standards, rather than "and." While these criteria are not absolute, they are widely accepted, as demonstrated in the NCBI classification system. The ANI values from the three websites were 94.53% (from orthoANIu), 94.3% (Kostas Lab), and 93.58% (JSpecies). From these values, the average (± SD) was 94.14% ± 0.50%. The only platform capable of calculating dDDH values is the Type (Strain) Genome Server, which does not provide SD values. Accordingly, this content has been revised in both the abstract and the main text (p. 1 lines 12-13; p. 3, lines 125-129; p. 7 lines 262-264). In addition, a substantiated demonstration of TT6 as a novel species is described in p. 13, lines 498-510.

Regarding the classification of a strain as a new species based on ANI and dDDH values, a study on Streptomyces species delineation proposed that an ANI value of 96.7% or a dDDH value of 70% could serve as thresholds for species delineation (https://doi.org/10.3389/fmicb.2022.910277).

A different study on Rhizobium species reported an ANI value of 95.9% and a dDDH value of up to 64.7% between certain strains, which were still classified as separate species https://doi.org/10.3389/fmicb.2022.910277. This indicates that within certain genera, species delineation can occur even with ANI values below 96% and dDDH values below 70%.

Comments 1-2: In addition, the title of Figure 1 needs to be changed, since the authors use a variant of MLSA to create the tree. In this case, "Phylogenomic tree based on whole genome sequences…" is incorrect. This is not a tree based on ANI values.

Response 1-2: Thank you for pointing out this. We agree with this comment. The original Figure legend has been changed as follows:

(page 6) Figure 1. Phylogenomic tree based on a total of 431 genes of strains TT6T, S. rosea M1T, S. mucosa 8-14-6T, and other closely related type strains. The NCBI GenBank accession numbers for each strain are shown in brackets. The RAxML method from PATRIC was used for gene comparisons. Elstera cyanobacteriorum TH019T was used as the outgroup. The bar indicates sequence divergence.

Comments 2-1: The authors' answers regarding drought tolerance or Triton X-100 degradation should be reflected in some form in the Discussion and Conclusion sections, without hindering further work by the authors on these topics. Without such additions, the manuscript appears logically incomplete on these issues.

Response 2-1: Thank you for pointing out this. We agree with this comment. The suggestion has been changed as follows:

a. The phrase ‘drought tolerance’ was added to the conclusion section (p 16. Line 582).

b. We added the results of Triton X-100 metabolism in the revision. Thus, the subtitle has been changed as 3.2.4. Genomic Analysis of Hydrogen, Sulfur and Triton X-100 Metabolism (p. 12 lines, 448) and; (p. 13, lines 471-476) The next content was added. “Our preliminary experiment using LC/MS analysis revealed that strain TT6T preferentially degrades higher molecular weight Triton X-100 by targeting the ethoxy chains, resulting in the accumulation of carboxylate intermediates (manuscript in preparation). It is hypothesized that dehydrogenases may play a role in this process. However, enzymes with specific dehydrogenase domain groups have not yet been identified in current studies, making it challenging to pinpoint the associated genes through genome analysis.”

Comments 2-2: I also see some inconsistencies in the manuscript. The authors discuss trehalose biosynthesis genes and the possibility of strain TT6 being drought tolerant, but Tables 2 and S8 show this strain as being unable to metabolize trehalose.

Response 2-2: Thank you for pointing out this. We agree with this comment. Our explanation is as follows:

We agree that the discrepancy between trehalose biosynthesis genes and the inability to metabolize trehalose requires clarification. This can explained: Synthesized trehalose may be metabolized via undetected pathways, secreted extracellularly, or diluted during cell division. Additionally, trehalose biosynthesis may primarily serve a protective role, such as aiding drought tolerance, rather than being metabolized.

Comments 2-3: Also, Table S9 shows that strain TT6 is α-glucosidase positive, but Tables 2, S7, and S8 show that this strain does not metabolize maltose and starch. I find these inconsistencies should be discussed by the authors in the manuscript.

Response 2-3: Thank you for pointing out this. We agree with this comment. The suggestion has been added as follows:

(p. 16, lines 263-264) The substrate is 6-Br-2-naphthyl-β-D-galactopyranoside, which exhibits a preference for aromatic aglycones. Consequently, maltose and starch in this table could test negative for metabolism.

Reviewer 2 Report

Comments and Suggestions for Authors

The manuscript has been significantly improved, and the authors have effectively addressed the previous concerns. It is now suitable for publication.

Author Response

Comments: The manuscript has been significantly improved, and the authors have effectively addressed the previous concerns. It is now suitable for publication.

Response: Thank you sincerely for your thorough review.